# Cyclic Nucleotide (cNMP) Analogues: Past, Present and Future

**DOI:** 10.3390/ijms222312879

**Published:** 2021-11-28

**Authors:** Erik Maronde

**Affiliations:** Institute for Anatomy II, Faculty of Medicine, Goethe-University, Theodor-Stern-Kai-7, 60590 Frankfurt, Germany; e.maronde@em.uni-frankfurt.de

**Keywords:** cyclic nucleotide analogues, cAMP, cGMP, 2′,3′-cGAMP, cyclic nucleotide binding proteins

## Abstract

Cyclic nucleotides are important second messengers involved in cellular events, and analogues of this type of molecules are promising drug candidates. Some cyclic nucleotide analogues have become standard tools for the investigation of biochemical and physiological signal transduction pathways, such as the *Rp*-diastereomers of adenosine and guanosine 3′,5′-cyclic monophosphorothioate, which are competitive inhibitors of cAMP- and cGMP-dependent protein kinases. Next generation analogues exhibit a higher membrane permeability, increased resistance against degradation, and improved target specificity, or are caged or photoactivatable for fast and/or targeted cellular imaging. Novel specific nucleotide analogues activating or inhibiting cyclic nucleotide-dependent ion channels, EPAC/GEF proteins, and bacterial target molecules have been developed, opening new avenues for basic and applied research. This review provides an overview of the current state of the field, what can be expected in the future and some practical considerations for the use of cyclic nucleotide analogues in biological systems.

## 1. Introduction

Since the discovery of the first so-called second messenger molecules by Earl Sutherland and coworkers [1,2], namely, “3′,5′-cyclic adenosine monophosphate” (cAMP; Figure 1, compound **1**) and “3′,5′-cyclic guanosine monophosphate” (cGMP; Figure 1, compound **2**), vast knowledge to understand their chemistry, mechanisms of action and their physiological functions has been accumulated.

For the finding that cAMP and cGMP bind to and activate certain protein kinases (which are nowadays called cAMP- or cGMP-dependent protein kinases, PKA and PKG), Edwin Krebs and Edwin Fisher were honored by receiving the Nobel Prize for Medicine or Physiology in 1992 [3].

Soon after their discovery, it became evident that the membrane permeability of cAMP or cGMP is too low to pass through cellular plasma membranes [4,5], one line of research became the synthesis of variants of cyclic nucleotides (cyclic nucleotide analogues/cNMPs) that maintain their principal functions while becoming more cell membrane permeable and thus applicable for use in living systems [6]. Soon cNMPs modified at various positions, which made them more (or less) potent, more target specific, or, while maintaining the principal function, less toxic in vitro or in vivo and provided higher cell membrane permeability, became available [5]. Moreover, analogues of the non-canonical cyclic nucleotides cCMP/cUMP and 2′,3′-cAMP (Figure 1, compound **8**) have been made [7] and analyzed for their occurrence and potential biological role [8,9].

It should be noted that in this review, the term cyclic nucleotide analogue will be used for every compound that is composed of a cyclic structure with at least one nucleotide component. The classical cNMPs (cAMP, cGMP, cCMP and cUMP) are a sub-group of this molecule class. However, also 2′,3′-cAMP/GMP/CMP/UMP and the more recently discovered cyclic dinucleotides and other nucleotide containing cyclic structure belong to this group. Moreover, many of the chemical synthesis and construction principles developed for cNMPs have been adapted (if applicable) for other second messenger systems like the phosphoinositides enabling some of the most advanced physiological investigations [10].

More than half a century after the discovery of cAMP and synthesis of the first cNMP- and other second messenger analogues, these molecules still provide an expanding toolbox for research and lead to novel findings in various organisms including bacteria and viruses. Thus, modern chemistry, advanced cell physiological methods and high-resolution microscopy have made second messenger analogues valuable research tools. Together with recent developments in drug application and delivery techniques, second messenger analogues will soon start to enter wider clinical use.

### 1.1. PKA and PKG as Cyclic Nucleotide Binding Proteins/cNMP Analog Targets

In the field of classical cNMP analogues, Jon Miller and R.K. Robbins with their coworkers were certainly among the pacemakers [11,12]. These groups synthesized hundreds of cNMP analogues including some still widely used 8-substituted-cAMP and -cGMP analogues like 8-Bromo-cGMP and 8-Fluorescein-cGMP (Figure 1, compound **10**).

Moreover, substitutions and modifications in the phosphate ring were introduced early [13] and tested for biological activity [14]. In particular, the replacement of the axial or equatorial oxygen with sulfur resulting in the phosphorothioate modification of the 3′,5′-cyclic phosphate ring system with an *Sp*- (axial) and an *Rp*- (equatorial) diastereomer; Figure 1, compound **4** and **3**), rendered these molecules more resistant against hydrolysis by most cyclic nucleotide phosphodiesterases (PDE) [15,16,17]. Together with the subsequent discovery that the axial *Sp*-isomer of a sulfur-modified cAMP analog (*Sp*-cAMPS) was agonistic, whereas the equatorial *Rp*-isomer (*Rp*-cAMPS) acted antagonistically on PKA [17], this modification became an important tool to evaluate the participation of PKA activity in biological systems [12,18].

The introduction of acetoxymethyl esters (“AM-Ester”) on the free hydroxy groups in the phosphate ring was another crucial modification step. AM-Ester-modification enhanced the membrane permeability by neutralizing the negative charge. The AM-ester-modified analogue passes the plasma membrane and enters the cytoplasm. Once the AM-ester is located inside the cell, cytoplasmic esterases hydrolyze the AM-ester groups, and the molecule gets trapped inside. Some AM-ester modified molecules were shown to increase the potency up to a 100-fold [5,19]. Beside cNMPs, AM-Ester modifications were also successfully used for the synthesis of other phosphate-containing second messengers, e.g., the phosphoinositides, and applied for the analysis of their cellular physiology [10,20].

In molecules where the AM-ester modification did not lead to stably applicable products other modifications to increase cell entry have been described. One type of novel, differently modified and higher lipophilic *Rp*-cAMPS-prodrugs with an acetoxybenzyl ester (AB-ester) was recently used successfully for fast, reversible inhibition of cAMP-dependent processes in pancreatic beta cells [21].

Beside the aspects mentioned above (target selectivity, agonism/antagonism, membrane permeability, toxicity, etc.) another line of cNMP analogue synthesis has been the introduction of functional groups like the highly reactive azido group (-N3)) for photoaffinity-labelling [22]. Moreover, fluorescent dyes to enable imaging by fluorescence microscopy [23], or spacers attached to a gel matrix for the preparation of affinity columns have been synthesized [24]. The state of affinity-tagged and fluorescent analogues and their use will be outlined in more detail below.

### 1.2. Non-PKA/PKG Cyclic Nucleotide Binding Proteins

Protein kinases like PKA and PKG are not the only cNMP-binding proteins. Other highly relevant and long known protein families which bind (or hydrolyze or both) are the 3′,5′-cyclic phosphodiesterases (PDEs) [25] and the organic anion exchangers (OAE) [26,27]. Since PDEs hydrolyze cNMPs, inhibition of PDEs leads to an elevation of cAMP or cGMP if the basal rate of adenylyl cyclase (AC) activity (AC is the enzyme that converts ATP to cAMP) is high enough. In many cells and tissues, basal adenylate cyclase activity is between 1 and 10% of maximal (induced) activity and may be inhibited further by, e.g., interleukin1-β as has been shown in rat liver cells [18].

The PDE family consists of a diverse group of enzymes which interact with cNMPs (mostly cAMP or cGMP) comprising different binding affinities [25]. They hydrolyze with different efficiency, display very different cell and tissue expression pattern and are, if mutated or otherwise altered to multiple diseased states, so-called “PDE-Opathies” [28]. Selective inhibitors of PDEs can be efficient elevators of intracellular cNMP concentrations in tissues expressing the “right” PDEs [29]. Prominent examples of specific PDE effects that have reached clinical relevance are the selective PDE5 inhibitors for erectile disfunction [30], PDE3/4 inhibitors for “chronic obstructive pulmonary disease” (COPD) [31] and non-COPD respiratory disease [32].

The exportation of cAMP and cGMP from cells is a long-known phenomenon [27]. The protein class mostly responsible for this exportation is the organic anion exchangers (OAE) of the SLC22 family which consists of 13 functionally characterized human plasma membrane proteins each with 12 predicted α-helical transmembrane domains. The family comprises organic cation transporters (OCTs), organic zwitterion/cation transporters (OCTNs) and organic anion transporters (OATs). The transporters operate as (1) uniporters which mediate facilitated diffusion (OCTs and OCTNs), (2) anion exchangers (OAEs) and (3) Na^+^/zwitterion cotransporters (OCTNs). They participate in small intestinal absorption and hepatic and renal excretion of drugs, xenobiotics and endogenous compounds and perform homeostatic functions in brain and heart. Important endogenous substrates include monoamine neurotransmitters, l-carnitine, α-ketoglutarate, prostaglandins, urate and the second messenger molecules cAMP and cGMP [33]. OAE can be inhibited by the well-established drug probenecid. Probenecid interferes with the OAT of the kidney, which reclaims uric acid from the urine and returns it to the plasma [34]. If probenecid (which is also an organic acid) is present, the OAT binds preferentially to it (instead of to uric acid), preventing reabsorption of the uric acid. By this mechanism, probenecid enhances the retention of drugs also transported by OAE and elevates their effectivity.

In cellular systems or tissues which express both PDEs and OAEs, inhibition of one or both can drastically increase the intracellular concentrations of cAMP and/or cGMP by inhibiting the degradation (PDE) and the export (OAEs) of these second messengers from the cell in which they are produced. Both systems have to be considered as potentially important factors modulating the action of externally applied cNMP analogues [29].

Beside the potential cNMP target protein families PDE and OAE mentioned above, “cyclic nucleotide-gated ion channels” (CNG), “hyperpolarization and cyclic nucleotide activated ion channels” (HCN) [35,36,37], cAMP-activated G-protein like proteins like “exchange factor directly activated by cAMP” (Epac) also known as “cAMP-regulated guanine nucleotide exchange factor” (cAMP-GEF) [35,36], cGMP transporters from the “multiple drug resistance receptor” class of proteins (MRP4 and 5) [37,38] and the “popeye domain containing (POPDC) gene family” [39,40,41] have been described as other cyclic nucleotide binding effector molecules with therapeutic perspective.

CNG channels have been first identified in the 1980s [42] and are a family of cNMP-binding and gated proteins that belong to the superfamily of voltage-gated potassium channels. Although belonging genetically into that family, CNG channels are virtually voltage independent [43]. The best-studied types are found in photoreceptor cells of the retina (CNG_ret_) and olfactory sensory neurons (CNG_olf_). There, CNG channels are gated by the second messengers of the visual and olfactory signaling cascades, cGMP and cAMP, respectively, and operate as transduction channels generating stimulus-induced receptor potentials. In visual and olfactory sensory cells, CNG channels conduct cationic currents to which calcium can contribute a large fraction, and calcium influx serves a modulatory role in CNG-channel mediated signal transduction. The search for selective cNMP analogues for CNG has led to the identification of 7-[2-(Phenylsulfonyl)ethyl]-8-thioguanosine-3′,5′-cyclic-monophosphate/7-PS(O)_2_;E-8-T-cGMP (Figure 1, compound **9**).

HCN channels were first described about 20 years ago. Thus far four isoforms are known (HCN1-4) which are expressed in heart and brain tissue. HCN2 is the main isoform in the brain, and HCN4 the main isoform in the sinoatrial node of the heart. Like the CNG channels, they belong to the superfamily of voltage-gated potassium channels, but unlike CNG, they are voltage-dependent. Mapping of the cNMP binding site has lead, among others, to 8-(2-[Fluoresceinyl]aminoethylthio)guanosine- 3′,5′-cyclic monophosphate (8-[Fluo]-cGMP/8-[Fluo]-AET-cGMP as a selective activator (Figure 1, compound **10**) [44].

cAMP also directly regulates Epac1 and Epac2, guanine nucleotide-exchange factors (GEFs) for the small GTPases Rap1 and Rap2 [35,36]. To selectively activate Epac in the presence of PKA, cNMP analogues were designed and tested for their ability to activate Epac while leaving PKA (or PKG) unaffected (“Epac activator”, Figure 1, compound **5**) [45]. Novel fluorescent Epac activators have been successfully used to distinguish effects in cardiac myocytes [46]. In the hippocampus formation of the brain, where Epac and PKA coexist, Epac activator and PKA-selective cNMPs have been shown to be able to separate their different influence on memory retrieval [47].

For these novel cNMP-binding proteins, whole new series of analogues were designed, synthesized and tested to be able to modulate one specific target while not affecting other common target molecules like PKA [48,49]. With the increasing number of cyclic nucleotide binding proteins, partially expressed in the same tissues in parallel [47], it becomes increasingly difficult to design selective analogues. Butt and coworkers have therefore determined the binding affinities for some widely used cyclic nucleotide analogues to a selection of physiologically relevant target proteins [50]. For example, the commonly used PKA activator, 8-BrcAMP, is also an efficient activator of Epac and was hydrolyzed by all PDEs tested (PDE1b, 2, 4, 5 and 10). It should also be noted that any analog that is a PDE substrate can also act as a competitive PDE inhibitor when present in large excess, as is the case in nearly all cell-culture studies [50]. An analysis of which cyclic nucleotide-binding (and/or hydrolyzing) proteins are expressed in a certain cell or tissue may therefore be necessary to find the most specific cNMP analog for the experimental purpose.

### 1.3. 2′,3′-cAMP and Other 2′,3′-cNMPs

Roughly a decade ago, experiments in the mouse kidney demonstrated that a second HPLC signal with a nearly identical retention time to 3′,5′-cAMP was actually identified as the positional isomer 2′,3′-cAMP (Figure 1, compound **11**) [9]. Soon thereafter, teams reported the detection of 2′,3′-cAMP and other 2′,3′-cNMPs (2′,3′-cGMP, 2′,3′-cCMP, and 2′,3′-cUMP) in biological systems ranging from bacteria to plants to animals to humans [51]. Injury appears to be the major stimulus for the release of these unique noncanonical cNMPs, which likely are formed by the breakdown of RNA. In mammalian cells in culture, in intact rat and mouse kidneys, and in mouse brains in vivo, 2′,3′-cAMP is metabolized to 2′-AMP and 3′-AMP; and these AMPs are subsequently converted to adenosine. Recent evidence points to a role of 2′,3′-cAMP in the release/secretion of exosomes by cells treated or not with sodium iodoacetate (IAA; glycolysis inhibitor) plus 2,4-dinitrophenol (DNP; oxidative phosphorylation inhibitor) [8]. A cell membrane-permeable form of 2′,3′-cAMP and 3′-AMP mimicked the potentiating effects of IAA/DNP on exosome secretion. In cells lacking 2′,3′-cyclic nucleotide 3′-phosphodiesterase (CNPase; an enzyme that metabolizes 2′,3′-cAMP into 2′- and 3′-AMP and serves as a marker for the oligodendrocyte glia cell type of the central nervous system [52]), effects of IAA/DNP on exosome secretion were enhanced [8].

### 1.4. Cyclic Guanosine Monophosphate-Adenosine Monophosphate (cGAMP) Synthase (cGAS)

Another potentially important and recently discovered cyclic nucleotide messenger system consists of the cytosolic DNA sensor protein “cyclic guanosine monophosphate-adenosine monophosphate” (cGAMP) synthase (cGAS). cGAS detects DNA and mediates downstream immune responses through the protein “stimulator of interferon genes” (STING, also known as MITA, MPYS, ERIS and TMEM173) [53].

Analysis of this messenger system showed that, compared to mouse STING, human STING shows greater preference for 2′,3′-cGAMP (Figure 1, compound **6**) than for the longer known molecules 3′,3′-cGAMP or c-di-GMP [54,55]. Together with the detection of 3′,2′-cGAMP in the fruit fly (*Drosophila melanogaster*), these findings suggest that such signal molecules may be present in many multicellular organisms including mammals [56].

### 1.5. Eukaryotic Pathogen (Cyclic) Nucleotide Binding Proteins

Specializations in the structure of cNMP-dependent signaling pathway proteins may provide an approach to target clinically relevant eucaryotic pathogens like *Plasmodium falciparum* [57], *Trypanosoma cruzi* [58] or others through design of cNMPs targeting the specialized protein structures in the pathogen while not affecting those of the host.

### 1.6. Prokaryotic Cyclic Nucleotide Binding Proteins

Many species of bacteria express cyclic nucleotide binding proteins, such as the transcription factor CAP, which binds to and is activated by cAMP [59]. More recently previously unknown nucleotide-based messengers have been discovered in various species of bacteria which utilize these to sense their environment and adapt [60,61].

Currently, known bacterial nucleotide messenger molecules include at least four categories [62]:3′–5′, 3′–5′ cyclic di-GMP regulates transitions between biofilm formation, motility, temperature sensation and many additional bacterial behaviors.3′–5′, 3′–5′ cyclic di-AMP controls osmoregulation in procaryotes.CD-NTases synthesize 3′–5′, 3′–5′ cyclic GMP-AMP, and many other cyclic-oligonucleotides (coNs), to induce phage (virus) defense.CRISPR/CAS III binding molecules [60,63].

Bacterial second messengers of the above-mentioned categories include such complex structures as cyclic mixed trimers and c-hexa-AMP (Figure 1, compound **8**) which have been identified recently [64,65,66,67,68,69,70,71,72,73,74]. It can be expected that there are many more interesting regulatory nucleotide molecules to discover in the future. An example for the dynamics of this part of the cyclic nucleotide field is the most recent finding that cCMP and cUMP, which had previously been detected in various mammalian tissues [75] mediate bacterial immunity against phages [76].

Due to the microbiomes in the human gastrointestinal system and other inner and outer spaces of the human body molecules of microbial origin like cCMP, cUMP and others already mentioned may turn out to be present in concentrations high enough to target the mammalian cNMP-binding proteins as well and have to be considered to be of physiological or pathological relevance in the host organs.

As stated above, many bacteria use the second messenger cyclic diguanylate (c-di-GMP) to control motility, biofilm production and virulence. Recently, a thermosensory diguanylate cyclase (TdcA) that modulates temperature-dependent motility, biofilm development and virulence in the opportunistic pathogen *Pseudomonas aeruginosa* was identified [77]. TdcA synthesizes c-di-GMP with catalytic rates that increase more than a hundred-fold over a ten-degree Celsius change. Analyses using protein chimeras indicated that heat-sensing is mediated by a thermosensitive Per-Arnt-SIM (PAS) domain similar to the eucaryotic PAS domain found in the “period” molecules Per1, Per2 and Per3 which play important roles in the eucaryotic circadian clock regulating “transcription-translation feedback loop” system [78]. In procaryotes TdcA homologs are widespread in sequence databases, and a distantly related, heterologously expressed homolog from the Betaproteobacteria order *Gallionellales* also displayed thermosensitive diguanylate cyclase activity. It was therefore proposed recently that thermotransduction may be a conserved function of c-di-GMP signaling networks and that thermosensitive catalysis of a second messenger constitutes a mechanism for thermal sensing in bacteria [77].

Hopefully, application of knowledge from the eukaryotic nucleotide signaling field into the bacterial/procaryote sensation field will provide new mechanistic insights and practical applications for the design of cNMP analogue structures in the future.

### 1.7. cNMP Analogues as Research Tools (Affinity Chromatography and Fluorescence Microscopy)

Beside their function as activators or inhibitors in physiological experiments, cNMP analogues have also been modified in order to be conjugated with different functional groups to attach them to gel matrix materials for affinity chromatography [24,79]. This method uses for example positions in the adenine ring or the free hydroxyl group (2′) in ribose and adds a “spacer” like aminoethylcarbamoyl immobilized to agarose. Such matrix material filled into chromatography columns is then loaded with extracts from cells or tissues of interest. The different cNMP-binding proteins attach to the nucleotide functional group in the agarose and can be fractionally eluted by perfusion with “free” (non-matrix-attached, soluble) cNMP. Using sequential elution with different cNMPs or concentration gradients of the free cNMP, binding proteins with different binding affinities to the immobilized affinity ligands may be eluted separately. These different eluant fractions can then be analyzed further by mass-spectrometrical analysis or classical amino-acid sequencing. Such “cNMP affinity chromatography” approaches were applied for the analysis of the cNMP-binding fraction of cells and made the analysis of cNMP-binding “subproteomes” feasible [24,79]. It can be speculated that miniaturized versions of such approaches (affinity techniques with mass-spectrometry) may allow “single cell proteomic” investigations in the near future.

With the advent of confocal and super resolution microscopic techniques, fluorescent cyclic nucleotide analogues like 8-Fluo-cGMP (Figure 1, compound **10**) and novel modified cNMPs with brighter and more stable fluorochromes have become another important line of cNMP modification [23,46]. The synthesis of brighter, less “bulky”, less (or more) photo-sensitive and “photo-activatable” fluorochromes in parallel with the improvement of high-resolution microscopical methods fuels the already impressive progress in this field [80,81].

### 1.8. cNMP Analogues in Pharmacological or Clinical Studies: Present and Future

To the best of my knowledge no classical cNMP analog has been authorized to market yet. Many of the potential reasons have been outlined above (and in the “Practical considerations for the use of cNMP analogues in biological systems” part below). However, Vighi et al. showed that liposomal encapsulation of cNMP analogues drastically enhances the half-life of the applied cNMP molecule in a biological system by slowing the fast egress from the cell [82] which partly is carried by the “organic anion exchangers” mentioned above. Thus, not only the modification of cNMP analogue structures but also the formulation in which they are applied is improving and will help reducing undesired effects of cNMP application.

Taken together the field of cyclic nucleotide and other second messenger analogues is both well established and active and has so far provided not only molecular and cellular tools but also candidates for pharmacological and clinical studies [82,83].

As of November 2021, 13,393 PubMed entries for cAMP analogues, 4866 for cGMP analogues, 1020 for *Rp*-cAMPS, 655 for *Sp*-cAMPS, 1005 for dibutyryl(db)-cAMP, 295 for 8-Br-cAMP and 306 for 8-**-cAMP (including 8-Fluorescein, 8-Chloro and different alkyl- or linker substituents) can be found. This, together with 89 PubMed entries for cAMP and cGMP analogue clinical trials [82,83], underscores the ongoing relevance of this over-60-years-old field for today and tomorrow.

## 2. Practical Considerations for the Use of cNMP Analogues in Biological Systems

Despite vast accumulated and confirmed knowledge about cNMPs and their abilities (including advantages and disadvantages), cNMP analogues are often used inappropriately or at least not in an optimal manner for the biological system of interest. Below is a list of points worth considering before using certain cNMP analogues to investigate (or modulate) cNMP-regulated pathways.

### 2.1. Cell Permeability/Membrane Passage

It is well established that unmodified cNMPs do not notably enter a cell if applied extracellularly [4]. Purine ring modified analogues (like 8-Br-cAMP or N6,O2′-dibutyryl-cAMP short db-cAMP) are more lipophilic and enter cells more efficiently but are often prone to degradation processes (see below). An extended analysis of the hydrophobicity (lipophilicity) of cNMPs and sources for the chromatographic retention parameters used for the determination of lipophilicity can be found in [84].

### 2.2. Metabolism/Degradation of the Analog

Many 8-substituted-cAMP analogues are degraded by phosphodiesterase (PDE) activity present in animal sera like fetal bovine serum which many cell culture systems contain [85]. The analogue metabolites may have unwanted activities like binding to adenosine receptors [86] or being transformed via the salvage pathway, thereby interfering with RNA and DNA metabolism. Although, for example, db-cAMP ideally hydrolyzes after cell entry into the butyrate molecules and mono- or non-butyrylated cAMP, the resulting butyrates may interfere with other cellular pathways. Moreover, db-cAMP may hydrolyze before entering the cell and release one or two butyrates thereby posing potential other, extracellular, side effects. On the other hand, in biological systems where butyrate and cAMP act in an additive manner (for example in nerve cell- or adipocyte-differentiation) this butyrate effect may be desirable. Under such conditions, db-cAMP acts as the “trojan horse” carrying several biologically active components into the cells.

In case the abovementioned effects and side effects are not desired, PDE-dependent hydrolysis by 3′,5′-cyclic phosphodiesterases can be avoided by using the PDE-resistant cNMP-phosphorothioate analogues like *Sp*- and *Rp*-cAMPS and their derivatives. Phosphorothioate analogues are also more lipophilic and thus enter cells easier. However, in the case of the presence of high-affinity adenosine receptors (K_D_ in the nanomolar range), it may be necessary to add the enzyme adenosine deaminase to the cell culture medium to convert the minute amounts (0.05%) of adenosine that may be present in *Rp*-cAMPS preparations to its inactive metabolite inosine [86].

### 2.3. Target Selectivity

As outlined above, cNMPs bind with high affinity to cAMP/cGMP-dependent protein kinases, 3′,5′-cyclic phosphodiesterases, EPAC/GEF proteins, ion channels (CNG; HCN), organic anion exchangers (OAG) and other target proteins. Each of the binding sites for cNMPs on these cNMP target molecules are different in terms of space, charge and flexibility so that in principle every modification that provides a gain of function on one of the target molecules may alter binding to another one thereby reducing the selectivity of the compound. An expression analysis of which cNMP target protein is present in a given biological system may help with the selection of the appropriate cNMP analog.

Taken together, the choice for the “ideal” cNMP analog for a certain experiment takes into account the nature of the investigated organism and/or the intended cell type targeted, combining good and fast permeability through the plasma membrane, stability against PDE hydrolysis and other enzymatic modifications outside and inside the cell with a high target selectivity and no or low toxicity.

## Figures and Tables

**Figure 1 ijms-22-12879-f001:**
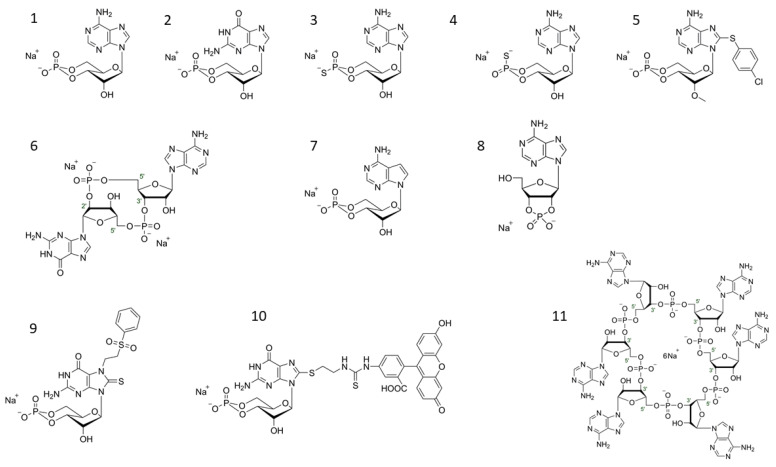
Selected structures of cyclic nucleotides (**1**) cAMP (Adenosine-3′,5′-cyclic monophosphate syn-conformation), (**2**) cGMP (Guanosine-3′,5′-cyclic monophosphate), (**3**) *Rp*-cAMPS (Adenosine-3′,5′-cyclic monophosphorothioate, *Rp*-isomer; PKA antagonist), (**4**) *Sp*-cAMPS (Adenosine-3′,5′-cyclic monophosphorothioate, *Sp*-isomer; PKA agonist), (**5**) 8-pCPT-2′-*O*-Me-cAMP (8-(4-Chlorophenylthio)-2′-*O*-methyladenosine-3′,5′-cyclic monophosphate; EPAC activator), (**6**) cGAMP(2′-5′)/2′3′-cGAMP/2′,5′-3′,5′-cGAMP (Cyclic (guanosine-(2′ − 5′)-monophosphate-adenosine-3′ − 5′)- monophosphate)STING ligand), (**7**) 7-CH-cAMP/cTuMP (7-Deazaadenosine-3′,5′-cyclic monophosphate; HCN ligand), (**8**) 2′,3′-cAMP (Adenosine-2′,3′-cyclic monophosphate), (**9**) 7-PS(O)_2_E-8-T-cGMP (7-[2-(Phenylsulfonyl)ethyl]-8-thioguanosine-3′,5′-cyclic monophosphate; selective CNG_olf/ret_ ligand), (**10**) 8-[Fluo]-cGMP/8-[Fluo]-AET-cGMP (8-(2-[Fluoresceinyl]aminoethylthio)guanosine-3′,5′-cyclic monophosphate PKG ligand, general cGMP-like binding fluorescence tool), (**11**) c-hexa-AMP; cyclic hexaadenylate (Cyclic hexa-adenosine monophosphate; CRISPR-CAS III ligand).

## Data Availability

Not applicable.

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
