# Peer review of "Cyclic Nucleotide (cNMP) Analogues: Past, Present and Future"

_ijms, 2021, doi:10.3390/ijms222312879_

Round 1

Reviewer 1 Report

A comprehensive review summarizing the newer research on cNMP effectors others than the classical kinases and phosphodiesterases.

Nevertheless I have some suggestions:

Page 2, first line: the example for 2`3`-cAMP/cGMP  (Figure 6) is not appropriate as it shows a dinucleotide mentioned later in the passage. I would suggest to omit Figure 4 (Sp) and show a 2´-3`-cyclic nucleotide.

Page 2, second last passage:  Some AM-ester

Page 2, last passage: …. Rp-cAMPS-prodrugs with an acetoxybenzyl ester (AB-ester) …

Page 3, second passage: adenylyl cyclase

Page 3, forth passage: the first sentence sounds extremely German. Better: The export of cAMP and cGMP from cells is a long known mechanism.

Page 4, last passage: that a second HPLC signal with nearly identical retention times to 3`-5`-cAMP was  actually identified as  the positional isomer 2`,3`-cAMP.

Page 6, first passage: c-hexa-AMP

Page 7, second passage: anion exchangers

Page 8, second passage: are there any data/paper about lipophilicity and cell permeability to look up for scientists?

Page 8, second passage: the author mentions the convertion of adenosine into Rp-cAMPS. What´s about the exchange of sulphur  (Rp-antagonist ) to oxygen resulting in an active compound?

Figure 1: Please start with the number and abbreviation followed by the full length name (easier to find).

Should´nt  cAMP be drawn in the  anti-conformation?

Reviewer 2 Report

This paper is well organized and comprehensively described with impressive insights into the field of cNMP analogues, which is of great importance for research and pharmacology. I recommend the paper to be accepted after some minor revisions.

I think the title is better to be changed into "cNMP analogues: Past, Present and Future", since the "cNMP analogues" is the focus of this paper and "second messenger" includes many other molecules like calcium, DAG, IP3, and etc.

Some mistakes should be corrected. For example,

Sone to Some in Page 2;

Several abbreviations should be explained at the first appearence, for example the Epac and COPD; 

I think it better to describe the CNG as "Cyclic nucleotide-gated ion channels" in page 3 since this is how the abbreviation comes (however, this is very minor);

At several positions, the eucaryotic and procaryotic should be eukaryotic and prokaryotic.  

In page 6, Pseudomonas aeruginosa should be Italic.
